# Begomovirus–Host Interactions: Viral Proteins Orchestrating Intra and Intercellular Transport of Viral DNA While Suppressing Host Defense Mechanisms

**DOI:** 10.3390/v15071593

**Published:** 2023-07-21

**Authors:** Sâmera S. Breves, Fredy A. Silva, Nívea C. Euclydes, Thainá F. F. Saia, James Jean-Baptiste, Eugenio R. Andrade Neto, Elizabeth P. B. Fontes

**Affiliations:** Department of Biochemistry and Molecular Biology/Bioagro, National Institute of Science and Technology in Plant-Pest Interactions, Universidade Federal de Viçosa, Viçosa 36570.000, MG, Brazil; samera.breves@ufv.br (S.S.B.); nivea.euclydes@ufv.br (N.C.E.); thaina_filli@hotmail.com (T.F.F.S.); james.baptiste@ufv.br (J.J.-B.); eugenio.neto@ufv.br (E.R.A.N.)

**Keywords:** begomoviruses, movement protein, nuclear shuttle protein, suppression of antiviral signaling, virus–host interactions, intracellular virus movement

## Abstract

Begomoviruses, which belong to the *Geminiviridae* family, are intracellular parasites transmitted by whiteflies to dicotyledonous plants thatsignificantly damage agronomically relevant crops. These nucleus-replicating DNA viruses move intracellularly from the nucleus to the cytoplasm and then, like other plant viruses, cause disease by spreading systemically throughout the plant. The transport proteins of begomoviruses play a crucial role in recruiting host components for the movement of viral DNA within and between cells, while exhibiting functions that suppress the host’s immune defense. Pioneering studies on species of the *Begomovirus* genus have identified specific viral transport proteins involved in intracellular transport, cell-to-cell movement, and systemic spread. Recent research has primarily focused on viral movement proteins and their interactions with the cellular host transport machinery, which has significantly expanded understanding on viral infection pathways. This review focuses on three components within this context: (i) the role of viral transport proteins, specifically movement proteins (MPs) and nuclear shuttle proteins (NSPs), (ii) their ability to recruit host factors for intra- and intercellular viral movement, and (iii) the suppression of antiviral immunity, with a particular emphasis on bipartite begomoviral movement proteins.

## 1. Introduction

Begomoviruses belong to the *Geminiviridae* family and exhibit monopartite or bipartite circular single-stranded DNA genomes [1]. Monopartite begomoviruses possess a single genomic component known as DNA-A and is approximately 2.7 kb in size, while bipartite begomoviruses consist of two components, DNA-A and DNA-B, with each measuring around 2.6 kb in size [1]. The successful establishment of a productive infection requires nuclear replication, cell-to-cell movement, and systemic transport throughout the plant via the phloem, which pose challenges due to the nuclear envelope and the cell wall [2]. The virion-sense strand of DNA-A (and monopartite genomes) encodes the coat protein (CP or AV1/V1), which may be involved in the intracellular transport of viral (v) DNA and insect transmission, and the movement protein (MP or AV2/V2), which is not present in all bipartite begomoviruses [1,3]. In monopartite begomoviruses, CP and MP play essential roles in inter- and intracellular movement of viral DNA (vDNA). The DNA-A complementary-sense strand encodes the replication-associated protein (Rep or AC1/C1), the transcriptional activator protein (TrAP or AC2/C2), the replication enhancer protein (Ren or AC3/C3), and the multifunctional AC4/C4 protein [1,3]. A fifth small open reading frame (ORF), which is designated AC5/C5 downstream of AC3/C3 in the DNA-A of some monopartite and bipartite begomoviruses, has been functionally characterized as a suppressor of RNA silencing defenses and facilitator of viral movement [4,5]. In bipartite begomoviruses, the DNA-B component encodes a nuclear shuttle protein (NSP) responsible for nucleocytoplasmic transport of vDNA and MP involved in cell-to-cell and systemic spread of vDNA [1,2]. The intra- and intercellular transport functions of begomoviral shuttling and movement proteins make them valuable targets for the identification of host factors in fundamental cellular processes exploited by begomoviruses. These viral proteins interact with host factors in various organelles and are therefore potential hubs in the host–virus protein–protein interaction network [6].

Understanding interaction networks is crucial for a comprehensive view of the metabolic and physiological responses of organisms to growth, cell differentiation, abiotic stress, and defense against pathogen attacks. Genome-wide studies focusing on plant immunity and pathogen infection strategies have provided an integrated picture of interactions between pathogens and plants [7]. This interaction network reveals that pathogen effectors and plant defense proteins converge, forming highly interconnected subsets of host proteins known as immune hubs [6,7]. Remarkably, these immune hubs conserve effectors from evolutionarily distinct pathogens, including the gram-negative bacterium *Pseudomonas syringae* (Psy) and the obligate biotrophic oomycete *Hyaloperonospora arabidopsidis* (Hpa) [7]. Conservation of some immune hubs include interactions with viral proteins. For instance, the CSN5A immune hub, which interacts with several host defense proteins and different effectors from divergent pathogens, also interacts with L2/C2 from begomoviruses [7,8]. A protein–protein interaction (PPI) network, derived from host protein interactions with the bipartite begomoviral transport proteins, MP and NSP, has been recently described [6]. The MP-NSP-host PPI network identified several hubs enriched by host transport functions and defense proteins, indicating that MP and NSP recruit host transport functions to move vDNA intra- and intercellularly and display immune-suppressing functions that favor infection. This review describes current knowledge about the role of viral movement proteins, potential target host factors involved in viral movement, and the mechanism of suppressing antiviral immunity by begomoviral movement proteins.

## 2. Nuclear Shuttling Properties of Begomoviral Proteins

After replication of infected cells in the nucleus, newly synthesized vDNA is transported to the cytoplasm through nuclear pores and subsequently to neighboring uninfected cells via plasmodesmata [9,10]. This crucial movement of vDNA from the nucleus to the cytoplasm is facilitated by viral proteins with nuclear-shuttling properties. These proteins were initially identified through molecular cell biology studies, focusing on fluorescently labeled viral protein and vDNA localizations [9,10,11]. Pioneering studies demonstrated that NSPs from different bipartite begomoviruses, such as bean dwarf mosaic virus (BDMV) and squash leaf curl virus (SLCV), exhibited nuclear localization, yet its subcellular distribution depended on co-expression partners [9,12]. As a nuclear shuttle protein, NSP tends to accumulate in the nucleus when expressed alone. However, during infection or in the presence of MP, NSP is directed toward the cell periphery [9,11,13]. The MP-mediated subcellular redirection is also a shared property of abutilon mosaic virus (AbMV)- and tomato golden mosaic virus (TGMV)-encoded NSPs [14,15]. Furthermore, NSP has been shown to interact with MP in the cytosol, which may serve as a driving force for the exit of vDNA complexes from the nucleus [16].

Consistent with the shuttling properties, NSP from SLCV harbors functional nuclear localization (NLS) and nuclear export (NES) signals, which may also apply to NSPs from other begomoviruses [17]. Furthermore, BDMV NSP has been demonstrated to transport labeled vDNA from the nucleus to the cytosol in the presence of MP [9]. Studies have shown that the optimized expression of AbMV NSP and MP can complement the intra- and intercellular movement functions of the bean yellow dwarf virus (BeYDV) mastervirus-derived replicon, providing evidence that the NSP nuclear-shuttling properties are conserved [18]. Accordingly, NSP interacts in vitro with single-stranded vDNA and double-stranded vDNA in a size-specific manner, although the extent and properties of these interactions may vary according to the virus-encoded NSP [12,19,20]. Apart from the presence of nuclear export sequences (NESs) that may enable NSPs to engage with the host nuclear export machinery, interactions between NSPs and exportins or the nuclear pore complex in the host cells have yet to be identified.

In monopartite begomoviruses, the function of NSP is taken over by CP, which is assisted by other viral proteins such as V2 and C4 [3]. Like NSP in bipartite begomoviruses, CP from various monopartite begomoviruses has been found to possess NLS and NES. It localizes to the nucleus and nucleolus and acts as a nuclear shuttle, facilitating the nuclear import and export of DNA [21,22]. CP interacts with ssDNA and dsDNA forms of the begomovirus genome via its N-terminal domain. CP from TYLCV and mungbean yellow mosaic virus (MYMV) also interacts with host proteins involved in nucleocytoplasmic trafficking, such as importin α and karyopherin α1 [23,24]. The TYLCV-V2 protein facilitates the nuclear export of V1 (CP) protein in a host exportin-α-dependent manner [25]. The V2-mediated export of V1/CP also depends on a specific V1–V2 interaction and has been shown to be crucial for viral spread and systemic infection. Additionally, a recently discovered small ORF from TYLCV called C5 has been identified to facilitate the movement of vDNA within and between cells [5].

## 3. Intracellular Transport of Begomoviral DNA

To facilitate the intracellular transport of vDNA, bipartite begomoviruses employ NSP, which hijacks the host transport active system and regulatory proteins to modulate its biochemical properties and function. While associated with vDNA, NSP recruits the acetylase nuclear shuttle interactor (NSI), forming a ternary complex that acetylates CP of cabbage leaf curl virus (CabLCV) [26]. This acetylation event causes CP to dissociate from vDNA within the nucleus of infected cells, thereby promoting the interaction between vDNA and NSP and facilitating the transport of the complex to the cytosol. Consistent with this mechanism, the overexpression of NSI has been shown to enhance the infectivity of CabLCV in *Arabidopsis thaliana* lines, indicating a proviral role of NSI [26].

Further evidence supporting this model comes from studies involving mutant NSI proteins [27]. These studies have revealed that the interaction between NSP and NSI is mediated by a 38-amino acid peptide (residues 150 to 187) within the viral protein. Mutations within this peptide, such as E150G, I164T, and D187G, disrupt the NSP–NSI interaction without affecting other NSP properties. CabLCV mutants carrying these mutations display reduced infection rates, lower vDNA accumulation, and attenuated symptoms compared to the wild-type CabLCV infection in Arabidopsis. Additionally, NSP interaction with NSI interferes with the oligomerization of NSI and reduces its assembly into highly active complexes, thereby partially compromising the NSI natural acetylation activity. Although NSI can still acetylate CP, its efficiency is lower compared to its natural targets [28]. The NSI expression pattern, predominant in young and/or sink tissues and vessels, suggests that NSP interference with NSI function in plant development may facilitate systemic viral infection [28].

In addition to acetylating CP, NSI is also able to acetylate core histones H3 and H2A, which interact with the phosphate backbone of vDNA to form minichromosomes [26,29,30]. H3 interacts simultaneously with vDNA and the movement proteins NSP and MP and may form a movement-competent complex [31]. In this complex, the H3 packaging property assists the transport of the viral genome through the nucleoporin complex and subsequently through plasmodesmata. Therefore, the NSI-mediated acetylation of histone H3 could impact the dynamic interaction between NSP and H3, serving as a critical posttranslational modification during begomovirus infection [26,31]. However, the pleiotropic effects of manipulating core histone levels pose a challenge in reverse genetics studies and hence a comprehensive biological understanding of their interaction with NSP is missing.

The precise mechanism underlying the efficient NSP-mediated transport of newly synthesized vDNA from the nucleus to the cytoplasm has yet to be fully understood. The specific interactions between viral NSP and exportins or the nuclear pore complexes remain unidentified. However, studies have revealed that NSP interacts with a cytosolic NSP-interacting GTPase (NIG), which localizes around the nuclear envelope and potentially assists the release of the NSP-vDNA complex from the nuclear pores into the cytosol [16,32]. Several lines of evidence support the proviral function of NIG in directing the NSP-vDNA complex to the cytoplasm. Firstly, NIG shares structural conservation and biochemical properties with the HIV Rev-interacting protein (hRIP), which is essential for the release of HIV-1 RNAs from the nuclear periphery to the cytoplasm in mammals [33]. The role of HIV Rev in mammals is similar to that of begomovirus NSP in plants [34]. Secondly, NIGs proteins from different plant species have been shown to bind in vivo with NSPs from various bipartite begomoviruses and promote the translocation of CabLCV NSP to the cytosol in transient assays [16,32]. Thirdly, NIG overexpression in Arabidopsis increases susceptibility to CabLCV infection. Finally, during infection, plant cells have developed an antiviral mechanism to counteract the NIG proviral function [35]. CabLCV infection triggers the accumulation of the nuclear body-forming WW domain-containing Protein 1 (WWP1) host protein that, in turn, binds and sequesters NIG into nuclear bodies, impairing its proviral function associated with cytosolic localization [35]. The newly replicated vDNA interacts with and disrupts the WWP1 nuclear body, restoring the NIG cytosolic localization and hence its proviral function.

A recent discovery has unveiled a new partner of NSP, referred to as the endosomal NSP-Interacting Syntaxin 6 domain-interacting protein (NISP), which also interacts with NIG. This interaction sheds light on the intracytoplasmic movement of the vDNA-NSP-NIG nucleoprotein complex [6]. Findings indicate that NIG does not facilitate the disassembly of the vDNS-NSP complex in the cytosol. Instead, it likely regulates the subsequent transportation of the NSP-vDNA complex to endosomes through protein–protein interactions [6]. Accordingly, NISP binds to NSP and NIG in the cytosol and may form a ternary complex, as the interaction between NIG and NISP is enhanced by a viral infection and NSP. Furthermore, a complex comprising vDNA, NSP, NIG, and NISP is found in the endosomes. By employing direct immunoprecipitation of endosomal NISP from chemically cross-linked complexes, it is evident that vDNA is pre-associated with the protein complex in the endosomes. Consequently, NSP might coopt an intracytoplasmic route involving NIG and NISP to guide the vDNA-NSP complex toward the endosomes. Consistent with this model, NSP from AbMV has also been shown to locate within the nucleus and associated with early endosomes in the presence of MP [36].

Alternative intracellular routes for bipartite begomoviral DNA have also been described [37,38,39]. These intracytoplasmic pathways have been uncovered based on the specific interactions of MP from the bipartite begomovirus (see MP interactions). These alternative routes may be considered additional routes shared by different begomoviruses to increase the efficiency of the intracellular transport of vDNA. They are based on interactions between MP and microtubule-associated host factors and may share conserved features with intracytoplasmic translocation of monopartite begomoviral DNA [36,38]. Microfilaments have also been shown to associate with the small C5 protein from the monopartite begomoviruses [5].

In monopartite begomoviruses, after the replication of the viral genome within the nucleus of infected cells, the transport of the newly synthesized vDNA to the cytoplasm is facilitated by CP [21,22,40,41]. In addition to mediating the translocation of vDNA from the cytoplasm to the nucleus, CP can shuttle the newly replicated DNA back to the cytosol via nuclear pores. Nuclear import and export depend on the ssDNA binding to the N-terminal domain of the CP and interaction with the nuclear machinery [42,43,44]. The complex CP-vDNA must translocate towards the cell periphery for the CP/V2-mediated cell-to-cell movement and systemic spread of the virus [45]. Recent studies have revealed the involvement of a small ORF called C5 in various aspects of intracellular vDNA movement [5]. TYLCV-C5 has emerged as a facilitator in the transport of vDNA within and between cells and may provide an intracellular pathway for the vDNA through interactions with microfilaments.

The βC1 protein derived from the circular single-stranded satellite DNAβ of cotton leaf curl disease (CLCuD) and the tomato leaf curl virus (TLCV)-encoded C4 protein have been suggested to potentially assist in the intracellular transport of begomovirus DNA [46]. Specifically, the satellite DNA β can complement the movement functions of DNA-B from bipartite begomoviruses in an βC1-dependent manner [46]. Similarly, the TLCV-C4 protein can confer movement functions to DNA-A from bipartite begomoviruses even in the absence of its cognate DNA-B. While the intracellular movement mechanism of βC1 is not well-described, its intracellular distribution suggests a role in transporting DNA A from the nuclear replication site to the plasmodesmata exit sites of the infected cell.

## 4. The Immunomodulatory Properties of NSP

In addition to its canonical transport function, NSP interacts with host defense proteins in various compartments to evade the plant immune system. As a suppressor of host defenses, NSP from CabLCV triggers the upregulation of asymmetric leaves 2 (AS2) expression and interacts with AS2 within the nucleus [47]. Additionally, NSP facilitates the trafficking of AS2 from the nucleus to the cytoplasm. Once in the cytoplasm, AS2 is redirected to the processing bodies (PBs), activating the decapping 2 protein (DCP2) and accelerating the turnover rate of mRNA. This interplay leads to a decrease in small interfering RNAs (siRNAs) and hampers the process of posttranscriptional gene silencing (PTGS), ultimately favoring the accumulation of viral mRNA. Consistent with these findings, AS2 overexpression enhances the virulence of begomoviruses, whereas inactivation of the AS2 gene reduces the susceptibility of the host [47]. Therefore, NSP negatively controls RNA silencing by increasing the expression and activity of AS2, an innate suppressor of PTGS, thereby favoring virus infection.

Furthermore, CabLCV NSP directly interacts within the nucleus with jasmonate insensitive 1 (JIN1 or MYC2) to inhibit its transcriptional activity [48]. JIN1/MYC2 is a conserved interaction partner shared by βC1 of tomato yellow leaf curl China virus and CabLCV NSP proteins [48]. JIN1/MYC2 is a transactivation factor that upregulates genes involved in terpene syntheses, such as terpene synthase 10 (TPS10) and terpene synthase 4 (TPS04). Accordingly, CabLCV infections result in reduced terpene synthesis, which, in turn, enhances the performance of *Bemisia tabaci*, the vector of CabLCV. Terpenes possess insecticidal, repellent, and attractive properties for natural enemies, thus serving as an effective defense strategy against *B. tabaci* [48,49]. Further investigations are necessary to fully understand the role of NSP in vector–plant interactions and the complex synergistic relationship between the vector and the virus.

Ras-GAP SH3 domain-binding protein (G3BP) plays a critical role in the formation of stress granules (SGs) in mammals when subjected to various stress conditions, including oxidative stress, heat shock, and viral infections [50,51]. The assembly of SGs, facilitated by G3BP, acts as an antiviral mechanism that hampers both host and viral protein translation during viral infections. However, viruses have developed strategies to counteract this host defense mechanism by disrupting SGs. For instance, poliovirus utilizes its proteinase 3C to cleave G3BP [52], and the alphavirus SFV interacts with G3BP through the FGDF motif present in the carboxyterminal region of the nsP3 protein [51,53]. The discovery of the FGDF motif in viral and host proteins has prompted investigations into similar motifs in plant virus-encoded proteins and their hosts [53,54,55]. In the case of begomoviruses, a conserved (F/Y) VS (F/Y) motif has been identified in the carboxyterminal region of the NSP protein, leading to the hypothesis that NSPs from AbMV and pea necrotic yellow dwarf virus (PNYDV) interact with the G3BP-like protein (AT5G48650) in plants. Accordingly, FVSF motif-containing AbMV NSP has been shown to interact with the G3BP-like protein [55]. Furthermore, a loss-of-function mutant (AASF) of AbMV NSP failed to induce SG assembly [55]. Despite the established interaction between begomoviral NSP and the G3BP-like protein, its biological significance and implications remain unclear.

At the plasma membrane, NSP has been shown to interact with two different classes of receptor-like kinases involved in antiviral plant immunity. These classes include the leucine-rich-repeat receptor-like kinases (LRR-RLKs), such as NSP-interacting kinase 1 (NIK1), NIK2, flagellin sensitive 2 (FLS2), and brassinosteroid insensitive 1 (BRI1)-associated kinase 1 (BAK1) [56]. The second class comprises PERK-like kinases, with NSP-associated kinase (NsAK) as a representative example [57].

NSP proteins from various begomoviruses interact with NIKs from distinct plant species, including *Solanum lycopersicum*, *Glycine max*, and *A. thaliana* [58,59]. Both CabLCV and TGMV NSP proteins interact with the activation loop of AtNIK1, preventing the phosphorylation of the activation site (Thr-474) and impairing its kinase activity. The binding site of NSP on NIK1 has been precisely mapped to an 80 amino acid stretch that overlaps with the activation loop and includes the crucial Thr474 residue [58]. Replacing the critical Thr-474 residue with aspartate generates a constitutively activated NIK1 mutant, which is no longer inhibited by NSP [60]. Therefore, the NSP inhibition of NIK1 kinase occurs upstream of Thr-474. The conservation of NSP–NIK interactions among different begomoviruses and NIKs from diverse plant species implies a mechanism wherein NSP negatively modulates host antiviral kinase receptors in an evolutionarily conserved manner.

In contrast to the inhibitory effect of NSP on NIK1-mediated antiviral immunity, begomoviruses-derived nucleic acids function as viral pathogen-associated molecular patterns (vPAMPs) and trigger the activation of the antiviral pathway [61,62]. NIK1 is activated by phosphorylation at the critical Thr-474 residue, which subsequently leads to the phosphorylation of the cytosolic Ribosomal Protein L10 at the Ser-104 residue (RPL10a) [15,63]. Once phosphorylated, RPL10a is redirected to the nucleus, where it interacts with the transcription factor LIMYB (L10-interacting MYB domain-containing protein) [64]. LIMYB is a critical repressor of genes associated with translational machinery and photosynthesis [62,64]. The prolonged activation of NIK1 leads to the suppression of global translation, which hinders the association of viral mRNA with polysomes, thereby impairing the efficient translation of viral mRNA and ultimately compromising infection [64]. NSP suppresses this host defense strategy by binding to the activation loop of the NIK1 kinase [58]. Ectopic expression of the T474D phosphomimic mutant, which bypasses NSP inhibition, enhances RPL10 translocation to the nucleus, and confers resistance to begomovirus infection in transgenic tomato lines [15,65]. Additional evidence suggests that NIK2, the closest homolog of NIK1, acts redundantly and is activated by begomovirus-derived nucleic acids, thereby initiating the assembly of the antiviral defense response, which is also negatively modulated by NSP [61,62].

Tomato yellow spot virus (ToYSV) NSP has been found to interact with *S. lycopersicum* (Sl)BAK1 (Solyc10g047140) through yeast two-hybrid assays [66]. Additionally, it interacts with the PERK-like kinase NsAK [57]. The formation of the NSP-NsAK complex enhances viral infection, suggesting a positive modulation role. In contrast, the interaction between NSP and BAK1 is likely associated with a host defense-suppressing function. BAK1 serves as a co-receptor for various pattern recognition receptors (PRRs), including flagellin sensing 2 (FLS2) and elongation factor-thermo unstable receptor (EFR), which detect specific PAMPs and trigger PAMP-triggered immunity (PTI), the initial layer of the plant’s innate immune system [67,68].

Multiple lines of evidence suggest that NSP may suppress BAK1’s role in immunity. Firstly, NIK1 and BAK1 belong to the same subfamily II of LRR-RLKs and share conserved positions for kinase activation sites [69]. Secondly, the binding site of NSP on NIK1 and the corresponding sequence on BAK1 are highly conserved [70]. Finally, NIK1-mediated antiviral signaling has been shown to interact with BAK1-mediated PTI through direct interaction of NIK1 with the co-receptor BAK1 and the PRR FLS2 [71]. The viral NSP suppressor of NIK1 may interfere with the formation of the NIK1-BAK1-FLS2 complex. Therefore, it is reasonable to hypothesize that the binding of NSP to BAK1 may also suppress PTI, although conclusive data supporting this interpretation are currently lacking.

## 5. Cell-to-Cell Movement Functions of Begomoviral Proteins

Plant viruses must move systemically within plants to induce disease. This process involves the intercellular movement of the viral nucleic acids from the initial site of infection to uninfected cells and subsequently spreading over long distances via the phloem. The cell-to-cell movement of plant viruses is facilitated by specific virus-encoded movement proteins [72]. These proteins effectively expand the size exclusion limit of plasmodesmata, allowing for active transport of viral genome to adjacent cells [72]. Among bipartite begomoviruses, the BDMV DNA-B-encoded movement protein (MP) was the first identified begomoviral protein exhibiting typical movement functions [9]. These pioneering studies have shown that the BDMV MP effectively increases the size exclusion limit of the plasmodesmata, thereby enabling efficient intercellular movement of the virus. Further studies have confirmed that MP orthologs from other bipartite begomoviruses, including SLCV, AbMV, and CabLCV, actively facilitate the cell-to-cell movement of vDNA. These studies found that loss-of-function mutants of MPs hindered the systemic spread of the virus but did not affect viral replication [73]. Gain-of-function studies have complemented these findings by demonstrating the ability of putative MPs to facilitate movement in otherwise movement-deficient virus mutants. Currently, it is conceptually accepted that the complementary- sense strand of the DNA-B component (BV1) encodes the typical movement protein (MP) from the bipartite begomoviruses.

Various mechanisms have been reported for targeting MPs to the plasmodesmata, and these mechanisms appear to be influenced by the interactions between MPs and host proteins. For instance, the MPs of CabLCV and SLCV have been found to interact with Synaptotagmin A (SYTA), a protein located in the endosomes of plant cells that regulates endocytosis [74]. SYTA has been demonstrated to be crucial for the cell-to-cell trafficking of different MPs [74,75]. These discoveries have led to the hypothesis that distinct viral MPs facilitate the intracytoplasmic transport of vDNA complexes to the plasmodesmata via the SYTA-mediated recapture endocytic pathway.

An alternative route for intracellular and intercellular movement of vDNA involves the formation of a stromule network following AbMV infection. The stromule network connects the nucleus and cytoplasm, extending towards the cell periphery and presumably anchoring at the plasmodesmata [39]. Stabilization of this network is facilitated by interactions with the plastid chaperone heat shock cognate 70 kDa protein (cpHSC70-1), which also interacts with MPs [37]. Consequently, it is plausible to assume that cpHsp70 acts as a docking site for bridging MP and viral nucleoprotein complexes to the stromule network for translocation toward the plasmodesmata, which potentially facilitates their spread to neighboring cells.

Several other proteins have been identified as potential partners of MP. These include histone (H3) [31], the peptidyl-prolyl cis-trans isomerase NIMA-interacting 4 (Pin4), and stomatal cytokinesis defective 2 (SCD2) [38]. Additionally, experimental studies conducted with AbMV infection have suggested the involvement of microtubules in the transit of MP towards the plasmodesmata, although a MP itself does not directly associate with microtubules in plant cells [38]. The association of MP with microtubules is likely facilitated by the microtubule-associated host factors Pin4 and SCD2 [38]. More recently, AbMV MP has been shown to be present in motile vesicles, trafficking along the endoplasmic reticulum in an actin-dependent manner towards the plasma membrane and plasmodesmata [36]. These findings underscore the complex and multifaceted mechanisms involved in the intercellular movement of vDNA, shedding light on intricate interplay between viral proteins, cellular components, and subcellular structures.

In the case of monopartite begomoviruses, the movement function has been attributed to the V2 protein encoded by the virion-sense strand of the single DNA component [21,76,77,78]. Evidence supporting the role of V2 as the movement protein includes mutational analysis, functional complementation assays using virus movement-defective mutants, the localization of V2 in plasmodesmata alongside C4, and interaction assays with host proteins involved in plasmodesmata regulation and viral movement [3,40,76,79]. However, variations may exist among different viral strains or species. This notion is highlighted by evidence suggesting that C4 may also possess movement functions [80,81,82,83]. Additionally, the movement of tomato yellow leaf curl virus-Israel (TYLCV-Is) does not rely on V2, as a totally loss-of-function V2 mutant retained a significant capacity to develop a systemic infection [84]. In contrast, mutations that abolish C4 expression also impair virus movement and systemic infection. The cell-to-cell movement functions in monopartite begomovirus infections are likely provided by two viral proteins, V2 (assigned as the movement protein, MP) and C4. However, both V2 and C4 are multifunctional proteins that also function as viral suppressors of RNA silencing. Therefore, the disruption of either the host defense-suppressing function or movement function of these viral proteins may render similar defective infection phenotypes, thereby hindering efforts to fully understand the mechanism by which V2 and C4 facilitate the intercellular transport of vDNA. A typical example has been observed in mutations within TYLCV-Is V2 [84]. Mutations that eliminate the RNA silencing suppression activity of V2 as well as total loss-of-function mutations result in a similar decrease in viral titer in infected plants. However, these mutations still retain a significant capacity to cause a systemic infection. These findings suggest that the reduced ability of the virus to proliferate is likely due to the loss of silencing-suppression activity in V2 rather than impairments in its movement functions.

Like the MP function observed in bipartite begomoviruses, which assists the nuclear export of the NSP-vDNA complex, the V2 protein interacts with the CP (also referred to as V1) within the nucleus of infected cells and facilitates the nuclear export of the vDNA-CP complex [3]. However, until recently, the mechanism by which V2 moves intracellularly toward the plasmodesmata remained elusive. Recent research has uncovered the function of the small ORF C5, providing a missing piece to this puzzle [5]. C5 has been shown to play a significant role in facilitating the movement of V2 toward the plasmodesmata. It associates with C2 in the nucleus and with V2 in the cytoplasm and at plasmodesmata. The interaction of V2 and C5 also facilitates their nuclear export and the relocation of V2 to the plasmodesmata. A separate small ORF V3 from TYLCV has been shown to move intracellularly along microfilaments to target the plasmodesmata [85]. The movement function of this small viral protein has been demonstrated through its cell non-autonomous localization and capacity to partially facilitate the traffic of a movement-deficient mutant of turnip mosaic virus (TuMV) into adjacent cells.

## 6. Functions of Movement Proteins in Suppression of Host Defenses

RNA silencing is a broad defense mechanism that plants employ against viral infections. To counteract this defense, the begomovirus genome encodes viral suppressors of RNA silencing (VSRs) [86,87]. Regarding begomoviral proteins associated with virus movement, V2 and C4 inhibit both transcriptional gene silencing (TGS) and posttranscriptional gene silencing (PTGS) [86,87,88]. V2 from cotton leaf curl Multan virus (CLCuMuV) effectively counteracts TGS by directly interacting with Argonaute 4 (AGO4) and suppressing RNA-dependent DNA Methylation (RdDM)-mediated TGS [89]. Similarly, TYLCV V2 interacts with host histone deacetylase 6, hindering the recruitment of Methyltransferase 1 (MET1) and resulting in hypomethylation of the viral genome [90]. TYLCV V2 is a potent suppressor of PTGS, interacting with and suppressor of gene silencing 3 (SGS3), the cofactor of the RNA-dependent RNA Polimerase 6 (RDR6), and inhibiting secondary siRNA production [91].

The C4 protein inhibits SAM synthetase enzyme activity and interacts with AGO4 to reduce viral genome methylation [92,93,94]. Additionally, specific C4 proteins interfere with the spread of the RNA silencing signal between cells and/or systemically [95,96]. TYLCV C4 interacts with plasma membrane-localized receptor-like kinases, barley any meristem (BAM) 1 and 2 proteins, preventing the intercellular spread of siRNAs [97]. C4 also suppresses kinases-mediated host defenses, including mitogen-activated protein kinases (MAPKs), which play crucial roles in general host defense against pathogens. Recent findings reveal that TLCYnV C4 inhibits MAPK-mediated defense responses by preventing the dissociation of the ERECTA/BKI1 complex [98]. In addition to BAM1 and 2, TYLCV C4 protein interacts with several other plant RLKs, such as clavata 1 (CLV1), FLS2, and BRI1 [99,100]. By interacting with CLV1, C4 disrupts the regulation of the antiviral factor WUSCHEL expression [101]. C4/AC4 has also been shown to interact with various shaggy-like protein kinases, which negatively regulate brassinosteroid (BR) signaling [102,103]. The interference of C4 with hormone signaling is further supported by its interactions with auxin biosynthetic enzymes, disrupting endogenous auxin content [104]. Notably, the NSP-MP-host protein–protein interaction network of bipartite begomoviruses revealed two significant hubs enriched with auxin response-related proteins and auxin signaling regulators [6]. These hubs consist of an NSP-MP general interaction-derived hub and an NSP-specific host interaction-derived hub [6]. Auxin signaling is likely targeted by the NSP and MP movement functions encoded by bipartite begomoviruses.

## 7. Models for the Intra and Intercellular Movement of Begomoviral DNA

Despite significant advancements in identifying molecular interactions between begomoviral movement proteins and host transport systems, the precise mechanisms underlying the intra- and intercellular transport of begomoviral DNA remain elusive. However, by piecing together the fragmented knowledge of the interaction network between viral movement proteins and host proteins, it is possible to construct comprehensive models that elucidate the nucleocytoplasmic, intracytoplasmic, and cell-to-cell movement of begomoviruses regarding the distinctions between monopartite and bipartite species.

For both monopartite and bipartite begomoviruses, the viral life cycle begins with the delivery of virions into the cytosol of plant cells by the *B. tabaci* vector (Figure 1 and Figure 2). Subsequently, the virions are unpacked. The CP-bound vDNA is then actively transported to the nucleus in a process facilitated by the interaction between CP and host nuclear import machinery, specifically importin α and karyopherin α1 [23,24,43]. Once inside the nucleus, the viral single-stranded DNA (ssDNA) is converted to double-stranded DNA (dsDNA) [1]. The dsDNA serves as a template for viral gene transcription and replication of the viral genome via the rolling circle mechanism.

For bipartite begomoviruses, NSP facilitates the translocation of the newly synthesized vDNA to the cytosol [56] (Figure 1). As a nucleocytoplasmic vDNA transport protein, NSP binds to vDNA, harbors NLS and NES, shuttles between nucleus and cytoplasm, and may interact with yet-to-be-identified nuclear/cytosolic receptors for nuclear import and export. In the nucleus, vDNA-associated NSP recruits the acetyltransferase NSI to acetylate, the viral CP substrate [26,27,28]. The acetylation of CP promotes its dissociation from vDNA, thereby favoring nuclear export over encapsidation of the vDNA. In addition, the histone H3 interacts with NSP and potentially protects the vDNA during the nuclear export [31]. On the cytosolic side, the GTPase NIG concentrates strategically around the nuclear pores to assist the release of vDNA-NSP complex from the nuclear pores into the cytosol [16]. NIG then guides the vDNA-NSP complex to the endosomes through interaction with the endosomal NISP, which also interacts with NSP [6]. The complex consisting of vDNA, NSP, NISP, and NIG is present in the endosomes. The movement protein (MP) interacts with the endocytosis regulator SYTA, causing its relocation from the plasma membrane to the endosomes [74]. This opportunistic relocation enables SYTA-associated MP to interact with vDNA-NSP at the endosomes. From the endosome, MP may utilize the SYTA-mediated recapture endocytic pathway to redirect the vDNA-NSP complex to the plasmodesmata and eventually to adjacent cells.

SYTA has also been shown to form ER-plasma membrane junctions that are subsequently relocated to plasmodesmata via its interaction with MP to regulate virus cell-to-cell movement [75]. The NSP-vDNA-NIG-NISP complex may also employ this alternative route by interacting with the microfilament network to dock at the plasmodesmata. The endosomal NISP may directly or indirectly attach viral complexes to both microtubules and microfilaments, as well as to SYTA-MP on early endosomes. While NISP does not associate directly with MP [6], the formation of a complex involving these proteins may be facilitated by the presence of vDNA, acting as a bridging factor.

The primary function of the MP is to increase the size exclusion limit of the plasmodesmata and actively transport the viral nucleic acid to neighboring cells [1]. Consequently, the movement functions of the MP are associated with its localization in the plasmodesmata. However, the MP has been found in several different compartments, including ER-derived tubes, microtubules, stromules, plasma membrane, endosomes, and nucleus, even though it only carries a predicted targeting signal to the cell periphery. The distribution of the MP to different cellular compartments is likely due to its association with specialized proteins, which may define alternative pathways to deliver the MP-nucleoprotein complexes to the plasmodesmata. In the case of AbMV infections, experimental data have revealed alternative routes for translocating the vDNA to the plasmodesmata based on the localization of the MP and its interaction with host proteins [36,37,38,39]. These data suggest that the vDNA employs a cytoplasmic pathway facilitated by the association of the AbMV movement protein (MP) with the endoplasmic reticulum (ER)-derived microtubules and motile vesicles. During this cytoplasmic route, the AbMV MP interacts with microfilament-associated proteins, which act as docking sites for translocating the MP-complex. These interactions with microfilaments support the movement of the MP toward the plasmodesmata, enabling the eventual transport of vDNA through these intercellular channels (Figure 1). Furthermore, a stromule network that is stabilized by Hsp70 and targets the AbMV MP has also been identified as an alternative cytoplasmic route for reaching the plasmodesmata [37,39]. These findings underscore the involvement of both microtubules and microfilaments in facilitating the intracellular movement of the vDNA during AbMV infections. Together, these cellular components and mechanisms contribute to the efficient spread of the virus through the plasmodesmata.

In monopartite begomoviruses, the nuclear-shuttling activity of NSP is replaced by CP, which is assisted by V2 and C5 proteins to promote the nuclear export of vDNA [3] (Figure 2). The interaction of V2 and C5 facilitates their nuclear export and the relocation of V2 to the plasmodesmata. Recent findings have shown that C5 interacts with microfilaments and promotes the redistribution of V2 to the plasmodesmata [5]. These interactions establish an intracytoplasmic pathway for the vDNA-CP-V2-C5 complex toward the plasmodesmata.

The ability of C5 to diffuse through the plasmodesmata to neighboring cells suggests that it may function as the movement protein responsible for actively transporting vDNA via the plasmodesmata [5]. However, the precise mechanisms and dynamics involved in vDNA trafficking through the plasmodesmata and the potential involvement of C5 or V2 in an active transport system are not yet fully understood. Nonetheless, the trafficking of the V2-C5 complex to adjacent cells provides a means to suppress host defenses prior to or concurrently with the plasmodesmata-mediated entry of vDNA into uninfected neighboring cells. This strategic movement of the complex may contribute to the successful evasion of host defense mechanisms.

## 8. Conclusions

Like all other viruses, begomoviruses must enter host cells and interact extensively with the host’s cellular machinery to complete their life cycle. Once inside the cytoplasm, the begomoviral genomes move intracellularly to the nucleus for replication and transcription of viral genes. Subsequently, the newly replicated vDNA must translocate back to the cytosol to dock at plasmodesmata for cell-to-cell movement. Many steps involved in the nucleocytoplasmic, intracytoplasmic, and cell-to-cell movement of vDNA are shared among both bipartite and monopartite begomoviruses. These shared features include the nuclear-shuttling property of NSP and the MP’s ability to increase the size exclusion limit of plasmodesmata. However, in monopartite begomoviruses, these movement functions may be accomplished by viral protein complexes rather than a single protein. For monopartite begomoviruses, the V2 protein is recognized as the movement protein, which requires not only host proteins but also the assistance of viral proteins to dock at the plasmodesmata.

For the intracytoplasmic movement, interactions between NSP–NIG and MP–SYTA are also conserved among NSPs and MPs from different bipartite begomoviruses. Additionally, the MP-mediated relocation of SYTA-induced ER-plasma membrane junction to the plasmodesmata is conserved among both monopartite and bipartite begomoviruses, even extending to RNA viruses. Despite significant progress in studying the interactions of begomoviral proteins with the host transport machinery in recent years, several questions regarding the mechanisms of intra- and intercellular translocation of vDNA remain unanswered. For instance, it is still unclear in what form (single-stranded or double-stranded DNA) the DNA leaves the nucleus and how the viral proteins with nuclear-shuttling properties engage with the host nuclear export machinery. Additionally, for monopartite begomoviruses, while in the nucleus, mechanisms underscoring the CP decision on vDNA packing or transport to the cytoplasm have yet to be elucidated. Furthermore, studies have identified both tubule-independent and tubule-guided mechanisms for the intracytoplasmic movement of vDNA, which depends on specific virus–host interactions. However, the decision-making mechanism behind the subversion of the preferential host transport system during infection remains unknown. Furthermore, the regulation of MP function during infection and interactions at the plasmodesmata for cell-to-cell movement are yet to be described. Uncovering these molecular mechanisms is crucial to develop more conceptual designs that will lead to alternative strategies in virus control. These studies will ultimately broaden our understanding of susceptibility genes. Should these genes exhibit characteristics of recessive resistance genes, wherein mutations are unlikely to affect host growth, the CRISPR/Cas technology could provide an efficient alternative for editing host susceptibility genes, an approach that holds the potential for achieving durable resistance.

## Figures and Tables

**Figure 1 viruses-15-01593-f001:**
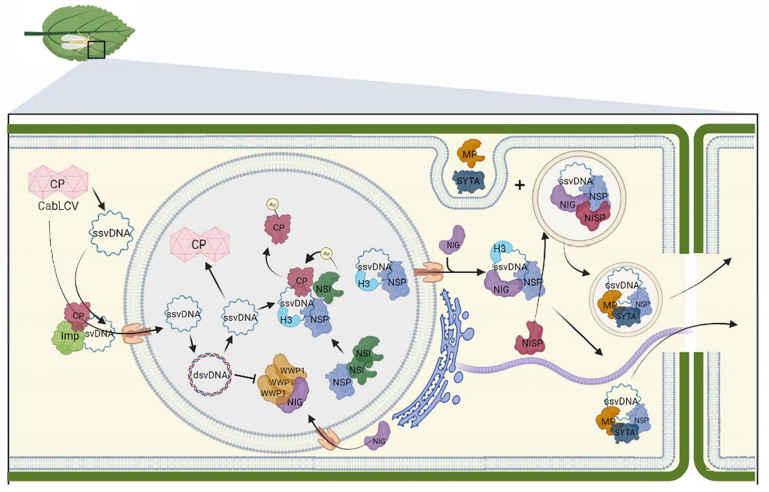
Current model of the intra- and intercellular movement of the bipartite begomovirus CabLCV. The viral particle is delivered into the cytoplasm by insect vectors upon cell infection. Importin-like proteins facilitate the translocation of disassembled CP-bound ssvDNA to the nucleus, where it is converted to dsDNA by host polymerases. Within the nucleus, NSP binds to both ssvDNA and dsvDNA, although the exact form of vDNA that is exported from the nucleus remains unclear. During this stage, CP binds to the synthesized ssvDNA for packaging. To prevent packaging, NSP recruits NSI, which acetylates CP, leading to its dissociation from vDNA. This process facilitates the NSP-mediated nuclear export of vDNA through nuclear pores. In the cytosol, NSP interacts with NIG, forming a complex that enables the release of vDNA-NSP from the nuclear pore into the cytosol. NSP and NIG further interact with the endosomal NISP, redirecting the vDNA-NIG-NSP complex to early endosomes. MP binds to the endocytosis regulator SYTA and is translocated from the plasma membrane to the endosome, where it interacts with the vDNA-NSP complex. From the endosome, the nucleoprotein complex utilizes the SYTA-mediated endocytic recycling pathway to travel to the plasmodesmata for the cell-to-cell vDNA-NSP movement. CabLCV MP can also relocate the SYPT-induced ER-PM junctions to the plasmodesmata for efficient cell-to-cell transport of vDNA. The vDNA-NSP-NIG-NISP complex may utilize this alternative route (secretory pathway along microtubules and microfilaments) for trafficking to the plasmodesmata. WWP1, an antiviral protein, sequesters NIG within nuclear bodies to prevent the NIG proviral function associated with its cytosolic localization. Nascent vDNA disrupts the WWP1 nuclear bodies restoring the NIG cytosolic localization and proviral function. The figure was created with BioRender.com (accessed on 27 June 2023) and adapted from [77].

**Figure 2 viruses-15-01593-f002:**
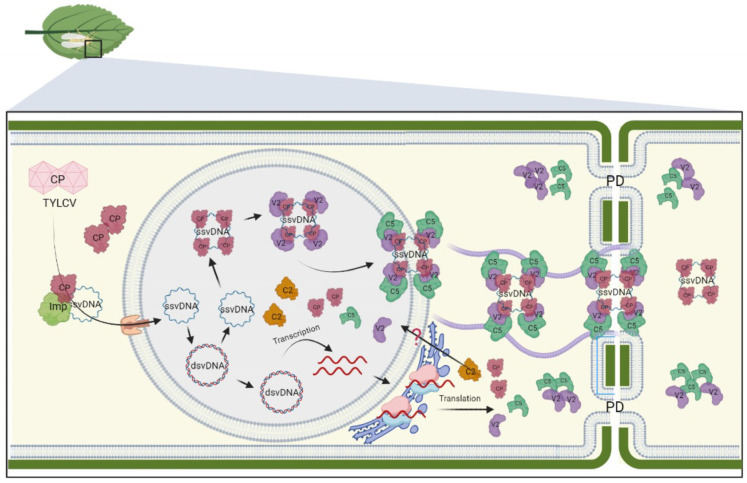
Current model for intra- and intercellular transport of monopartite begomoviruses. Upon cell infection, the viral particle is delivered into the cytoplasm by insect vectors. Importin-like proteins facilitate the translocation of disassembled CP-bound ssvDNA to the nucleus, where it undergoes conversion to dsDNA by host polymerases. CP, in association with V2, interacts with newly synthesized vDNA within the nucleus. The association of C5 with V2 enables the transport of the vDNA-CP-V2 complex from the nucleus to the cytosol. Once in the cytosol, the nucleoprotein complex-associated C5 interacts with microfilaments and guides the complex toward plasmodesmata to promote the viral cell-to-cell movement. Additionally, C5 can diffuse to adjacent cells through plasmodesmata, where it relocates the V2-C5 complex to suppress host defense mechanisms in neighboring cells. The figure was created with BioRender.com (accessed on 28 June 2023) and adapted from [5].

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
