# Peer review of "Begomovirus–Host Interactions: Viral Proteins Orchestrating Intra and Intercellular Transport of Viral DNA While Suppressing Host Defense Mechanisms"

_viruses, 2023, doi:10.3390/v15071593_

Round 1

Reviewer 1 Report

In this manuscript, the authors summarize the role of viral transport proteins, specifically movement proteins (MPs) and nuclear shuttle proteins (NSPs), and their ability to recruit host factors for intra- and intercellular viral movement. They also describe suppressive functions of host defenses by MPs. Finally, model for intra- and intercellular transport of monopartite begomoviruses or the bipartite begomovirus 440 CabLCV is proposed. The paper is well written and t I think it is suitable to be published after minor revision. 

 Minor issues: 

 The recent report indicates that V3 protein traffics along microfilaments to plasmodesmata to promote virus cell-to-cell movement, please include the information [ref: Pan Gong, Siwen Zhao, Hui Liu, Zhaoyang Chang, Fangfang Li, Xueping Zhou. 2022. Tomato yellow leaf curl virus V3 protein traffics along microfilaments to plasmodesmata to promote virus cell-to-cell movement. SCIENCE CHINA-Life Sciences 65:1046-1049]

Line166-168: “CabLCV infection triggers the accumulation of the nuclear body-forming WWP1 host protein that, in turn, binds and sequesters NIG into nuclear bodies, impairing its proviral function associated with cytosolic localization”. Please provide a full name of WWP1 at first use in the main text.

 Lines 521-524: “Nonetheless, the trafficking of the C2-C5 complex to adjacent cells provides…..”. V2-C5 complex? Please check it.

Lines 534-537: “Additionally, C5 can diffuse to adjacent cells through plasmodesmata, where it relocates the C5-C2 complex to suppress host defense mechanisms in neighboring cells. The figure was created with BioRender.com and adapted from [5]”. V2-C5 complex? Please check it.

 English Language is fine.

Author Response

Thank you very much for your comments on our manuscript, which certainly contributed to making it better. Therefore, we have accepted your suggestions as follows:

The significant literature regarding V3 (Gong et al., 2022) was included in the text. Please see lines 393-397

In lines 166-166,  the full name of WWP1 (WW domain-containing Protein 1) was included in the text.

In lines 529-530, the complex “C2-V5” was replaced with “V2-C5” (in red)

In lines 543-544, we also corrected the same mistake. Now see the complex V2-C5 in red.

Sincerely,

Elizabeth P B Fontes

Reviewer 2 Report

This is a well-written review of Begomovirus-host interaction, covering most state-of-the-art studies despite some minor grammar mistakes. In addition, I have two suggestions for the authors to consider. Firstly, it would be better to specify the virus and host names when giving specific examples. For instance, in line 219, NSP triggers the upregulation of AS2, ref 47. It sounds like all Begomovirus NSP can trigger AS2 upregulation;  In line 324, "MP orthologs from other bipartite begomoviruses", ref 73, Please give some virus names. In addition, per the models. In Figure 1, the ssvDNA directly points to CP in the nucleus. Does this mean the CP is synthesized in the nucleus or the Cp is packed into virions? If the latter scenario is the case, the "CP" should be replaced by "virion." In addition, the organelles should also be marked, such as "virus-induced vesicle," "plasmodesmata," etc.

Minor suggestions:

Line 37, delete the parenthesis. “([1,3]”

Line 45, delete the parenthesis. “([4,5]”

The scientific names should be in italics. Like Arabidopsis thaliana, Bemisia tabaci, etc.

Please align the format of the reference. Line 654

Title of 6, "The movement protein suppressive functions of host defenses," can be replaced by "Functions of movement proteins in suppression of host defenses"

 some minor grammar mistakes were found.

Author Response

We are very grateful to reviewer 2 for its helpful and constructive comments on our manuscript. We have endeavored to address all concerns of the reviewers, which have helped us strengthen our manuscript. Point-by-point responses are provided below each comment of the reviewers, and the corresponding modifications are highlighted (red) in the manuscript text.

Reviewer: This is a well-written review of Begomovirus-host interaction, covering most state-of-the-art studies despite some minor grammar mistakes.

Authors’response About the minor error mistakes. Hard to find them but we certainly tried.

Some of them:  inmate was replaced with innate (line 228)

See new written format to increase readability; line 222; lines 252-254

Reviwer: In addition, I have two suggestions for the authors to consider. Firstly, it would be better to specify the virus and host names when giving specific examples. For instance, in line 219, NSP triggers the upregulation of AS2, ref 47. It sounds like all Begomovirus NSP can trigger AS2 upregulation;  In line 324, "MP orthologs from other bipartite begomoviruses", ref 73, Please give some virus names. In addition, per the models.

Authors’ response: Excellent comment. Please, see now in the revised version (in red, line 219) the required specifications. We also revised the text for other missing virus names. See now in red: lines 108-109;  line 114; line 211; lines 252-253

Reviewer: In Figure 1, the ssvDNA directly points to CP in the nucleus. Does this mean the CP is synthesized in the nucleus or the Cp is packed into virions? If the latter scenario is the case, the "CP" should be replaced by "virion." In addition, the organelles should also be marked, such as "virus-induced vesicle," "plasmodesmata," etc.

Authors’ response: That's an excellent point. In our model, we believe (as is the case in most geminiviral studies) that the virion is uncoated in the cytosol. However, some CP is still bound to ssvDNA, facilitating the translocation of ssDNA to the nucleus through its interaction with importins. Furthermore, we are unsure whether the virus exits the nucleus as a virion or with CP-bound DNA. We have addressed these matters in the discussion section to draw attention to them.

 Minor suggestions:

Line 37, delete the parenthesis. “([1,3]”

OK

Line 45, delete the parenthesis. “([4,5]”

OK

The scientific names should be in italics. Like Arabidopsis thaliana, Bemisia tabaci, etc.

OK (see all scientific names in red)

Please align the format of the reference. Line 654

Thank you, OK

Title of 6, "The movement protein suppressive functions of host defenses," can be replaced by "Functions of movement proteins in suppression of host defenses"

 The original title of 6 was replaced with the one suggested by the reviewer.

Reviewer 3 Report

The review by Breves et al. is timely and well written. The manuscript presents a well-balanced summary on how geminiviral DNA moves intra- and intercellularly based on 30 years of research on this topic. The authors have done an excellent job in reviewing all the literature on this topic from the early 90s onwards. I very much appreciated reading this manuscript. In particular, the authors have carefully explained the differences between the various mechanisms by which the two viral groups, mono- and bipartite geminiviruses, modulate host functions like nuclear export, plasmodesmata, and plant immunity citing the work of many researchers.

Suggestions:

I missed a table that summarizes the different viral functions (rows):

- monopartite and biparite species (as columns), movement: MP or V2+C5; shuttling of vDDNA: NSP or CP, function, etc.

- It could include the discussed host proteins as well that interact with the different viral proteins.

This table will support the reader with guidance on the different functions discussed. Some of this information can be found in the Figures 1 and 2, but a Table could assist the reader as well.

In addition, section 7 appears to be repetitive as the information is presented above as well. Can it be integrated with an earlier part of the text or can it be integrated with the discussion?

Minor observations:

L125, L237, L239 L432: A. thaliana, Bemisia tabaci -> Italics; please check the organism names.

Please check the usage of Capitol letter for the plant protein names, e.g., L223 "Decapping.." versus L396 "argonaute 4". Other cases apply as well.

Author Response

We are very grateful to Reviewer 3 for their helpful and constructive comments on our manuscript. We have made every effort to address the the reviewer's concerns, which have greatly assisted us in strengthening our manuscript. Below each comment from the reviewers, you will find point-by-point responses along with the corresponding modifications highlighted in red within the manuscript text.

Suggestions:

Reviewer: I missed a table that summarizes the different viral functions (rows):- monopartite and bipartite species (as columns), movement: MP or V2+C5; shuttling of vDDNA: NSP or CP, function, etc.- It could include the discussed host proteins as well that interact with the different viral proteins.

Authors’ response: We appreciate the feedback and suggestions provided by the reviewers. In response to their recommendations, we made efforts to design a table for the original version of the review. However, we encountered challenges due to the varying establishment level of movement protein in different monopartite begomoviruses.

For instance, apparently, V2 does not function as a movement protein in TYLCV-Is, while C4 is believed to perform that role instead. We have thoroughly incorporated this discussion into the review to ensure accurate and up-to-date information is presented.

Furthermore, in response to reviewer 1's suggestion, we have included information about the potential role of C3 as a movement protein, along with supporting evidence. This addition enhances the comprehensiveness of the review and allows readers to delve deeper into the complexities of these viral systems.

While we understand the interest in including a table, a complete table may oversimplify the nuances and ongoing discussions in the field. We aim to present a balanced and thorough analysis, and a table could lead readers to perceive the information as absolute and potentially disregard the detailed discussion in the text.

In light of these considerations, we kindly request the reviewer to consider our argumentation when deciding on the necessity of a table. We value the reviewer's expertise and feedback and remain open to further discussion and suggestions to improve the presentation of the information to ensure both clarity and depth of understanding for the readers. 

 Reviewer: In addition, section 7 appears to be repetitive as the information is presented above as well. Can it be integrated with an earlier part of the text or can it be integrated with the discussion?

We agree with the reviewer's observation that Section 7 was somewhat repetitive. However, it was intentionally designed to consolidate fragmented information into a cohesive model. Therefore, we believe it would be appropriate to retain it in its current form.

Minor observations:

Reviewer: L125, L237, L239 L432: A. thaliana, Bemisia tabaci -> Italics; please check the organism names.
Authors’ response: Please, see now in red all scientific names in italic as recommended.

 Reviewer: Please check the usage of Capitol letter for the plant protein names, e.g., L223 "Decapping.." versus L396 "argonaute 4". Other cases apply as well.

Authors’ response: In scientific literature, it is common to use capital letters for the full names of plant proteins, such as "NUCLEAR SHUTTLE PROTEIN-INTERACTING KINASE 1." This convention is widely accepted and understood within plant-focused journals and scientific communities. However, it may be less familiar or recognized in general journals. So, we will adhere to the established practice of capitalizing the first letter of plant protein names to match the abbreviations throughout the manuscript to maintain consistency.

Please see in red all protein names that was changed according to this rule.

Reviewer 4 Report

The review article “Begomovirus-Host Interactions: Viral Proteins Orchestrating Intra and Intercellular Transport of Viral DNA while Suppressing Host Defense Mechanisms” by Sâmera et al focuses on how begomoviral MPs and NSPs regulate viral movement via modulation of host cellular factors and antiviral immunity. This is a well-written, clear and interesting review. I have a couple of suggestions for authors though:

1.      Authors should supplement the review by adding few lines along the idea about how mutations affect the ability of begomoviral proteins to transport viral DNA and/or suppress the host defences.

2.      What are the emerging trends regarding development of begomoviral management strategies through interruption of their inter/intra cellular movement proteins?

Author Response

We are very grateful to Reviewer 4 for their helpful and constructive comments on our manuscript. We have made every effort to address the the reviewer's concerns, which have greatly assisted us in strengthening our manuscript. Below each comment from the reviewers, you will find point-by-point responses along with the corresponding modifications highlighted in red within the manuscript text.

 Reviewer:  Authors should supplement the review by adding few lines along the idea about how mutations affect the ability of begomoviral proteins to transport viral DNA and/or suppress the host defences.

Authors’ response: Thank you for this suggestion which was incorporated in lines 379-383 (in red)

Reviewer: What are the emerging trends regarding development of begomoviral management strategies through interruption of their inter/intra cellular movement proteins?

Authors’ response: Please refer to the revised version (highlighted in red, in conclusion, lines 578-582) where we have included a potential strategy to disrupt virus movement as a means of developing resistance.

Sincerely,

Elizabeth P B Fontews